# The Effectiveness of Gamified Tools for Foreign Language Learning (FLL): A Systematic Review

**DOI:** 10.3390/bs13040331

**Published:** 2023-04-13

**Authors:** Zhanni Luo

**Affiliations:** School of Foreign Languages and Literatures, Chongqing Normal University, Chongqing 401331, China; zhanni.luo@cqnu.edu.cn

**Keywords:** gamification, foreign language learning (FLL), effectiveness, literature review

## Abstract

Gamification has emerged as a promising approach for foreign language learning (FLL), which refers to the use of game design elements to engage learners or improve academic performance. However, the features of gamification studies in FLL and their effectiveness are unclear. Additionally, how previous studies measured the effectiveness of gamified FLL tools is not well understood. In this systematic review, this author addressed these questions based on 21 empirical studies. The findings revealed that the effectiveness of gamified tools in FLL was mixed, with some bringing positive changes, others negative changes, and some showing no differences. The factors that influenced the effectiveness include methodological limitations, biases in the experiment setting, technical limitations, individual differences, failure to achieve meaningful gamification, a mixture of element selection, sub-optimal measurement, and data interpretation biases. This study identified research gaps in previous studies and offers suggestions for future research in this area.

## 1. Introduction

### 1.1. Background

In the last two decades, there has been a widespread discussion about technology-based or technology-enhanced language learning. The use of digital devices has revolutionized language learning by providing a range of technological conveniences. These include voice recording, material storage, automated speech recognition, and grammar checking [1,2,3], which have made language learning more efficient and effective. Moreover, technology has enhanced the visual aspect of learning by presenting materials in a more engaging and interactive manner [4,5]. This, coupled with rapid feedback, has enabled learners to receive immediate corrections to their work, leading to faster progress and mastery [4,5]. Furthermore, digital devices have made self-study and personalized learning more accessible, empowering learners to take control of their learning [1,6,7]. The use of bite-sized lessons has also made learning more manageable [8] and has provided a student-centered learning experience [1].

The introduction of portable digital devices, such as iPads and smartphones, has further expanded the possibilities for language learning. These devices are ubiquitous and portable, enabling learners to study at any time without being tied to a particular geographical location [3,7,9,10,11]. The ubiquity and portability are particularly important for foreign language learning (FLL), as FLL often involves dull, time-consuming, and perseverance-demanding practices, such as vocabulary memorization [11,12]. In-class time is typically limited, making it difficult for learners to devote adequate time to such persistence-demanding activities. As a result, learners need to utilize their fragmented time outside the classroom to improve their skills [13].

Despite the benefits of technology-based language learning, it is important to note that engagement maintenance is an issue. Technology-based language learning tends to be self-directed, which may result in a lack of synchronous interactions with peers or instructors [5,10,14,15]. As a result, researchers criticized the effectiveness of technology-based language learning, arguing that without real-time stimuli, learners may become disengaged and prefer amusement or rest [16]. In the long run, this disengagement can even lead to learners dropping the class [16,17]. Thus, maintaining learner engagement is a critical issue in technology-based language learning [11].

### 1.2. Research Gaps

Gamification, which is the use of game design elements in non-game contexts to promote the expected behaviors [18], shows promise in maintaining learner engagement.

However, researchers doubted whether it is effective at bringing pedagogical benefits. For example, participants in the study by Luo, Brown and O’Steen [19] expressed pessimism about the effectiveness of gamification-supported foreign language learning. They believed that while the concept of gamification is great, few products can achieve what they promised. [19], Luo [20] further noted that perceived usefulness is a critical factor in determining whether new educational technologies are accepted. Despite this, there have been few studies to date that specifically focused on the ***effectiveness*** of gamification in foreign language learning.

Furthermore, gamification is a concept that originated from video games, and thus, a considerable number of researchers are video game designers or information technology specialists. In other words, in the field of gamification-supported foreign language learning (FLL), FLL researchers are not the mainstream. Accordingly, researchers criticized educational gamification products for being poorly integrated with pedagogical contents, and their educational benefits can hardly be ensured [19,21]. This suggests that there is a need for collaboration between FLL researchers and gamification experts to ensure that gamified tools are designed to effectively support language learning while incorporating appropriate pedagogical principles.

## 2. Literature Review

### 2.1. Gamification: Definition and Importance

Gamification is the use of game design elements in non-game contexts to promote expected behaviors [18]. Gamified learning, then, is defined as the use of game design elements for educational purposes [21]. Gamification is a well-acknowledged concept for boosting engagement, and it has been regarded as promising in both computer-assisted language learning (CALL) and mobile-assisted language learning (MALL).

The classification of game design elements has been subject to debate. Luo [21] categorized game elements into two groups: explicit game design elements and implicit game design elements. Explicit game design elements refer to the obvious game-like elements that people can see in commercial video games, such as points, badges, leaderboards, avatars, and virtual currencies. Implicit game design elements refer to the underlying mechanisms that make gamification activities engaging, which are connected to individuals’ innate psychological needs. Implicit game design elements are abstract nouns, such as feedback, achievement, competition, collaboration, challenge, avoidance, ownership, and user control.

Notably, gamification can take both digital and non-digital forms. However, due to the challenge of describing and implementing non-digital gamified interventions, this study specifically focused on digital gamification. In this study, gamified learning tools refer to digital educational websites, information systems, or mobile applications that employ game design elements [19]. Furthermore, the gamified FLL tools in this study specifically refer to gamified tools designed for foreign language learning (as presented in Table 1).

Either for engagement enhancement or achievement improvement, gamification for foreign language learning has become a clear trend in recent years [2]. Purgina, Mozgovoy and Blake [2] stressed that foreign language learning requires long-time commitment, which involves numerous repetitive tasks and memory drills that hardly can be considered entertaining, and thus, any technological tricks that make this undertaking less daunting should be appreciated.

### 2.2. Research Focuses of Other Literature Reviews

For a literature review, appropriate research questions (RQs) are important. There is a lack of literature review studies that focus on this exact topic (the effectiveness of gamification for foreign language learning), and thus, this author referred to literature reviews on similar topics. The referred review articles were on the topic of digital game-based learning, the impact of gamification on education, and technologies for foreign language learning.

Hung, Chang and Yeh [22] conducted a review of digital game-based language learning research from five aspects: publication year and location, target language, learner background, the involved digital games, and research methods. The first four research aspects were summarized as substantive features by Xu, Chen, Eutsler, Geng and Kogut [23]. Therefore, the first research question was raised (RQ 1: What are the substantive features of the selected gamified FLL studies, including the involved tool, the target language, the target learning content, and the learner’s background?).

Previous review articles covered methodological features to a large extent. In particular, All, Castellar and Van Looy [24] emphasized the importance of effectiveness measurements in digital game-based learning, which included five dimensions of methodological analysis: participants, intervention, methods, outcome measures, and data analysis methods. Since there is a need for investigations of outcome measures, this author divided the methodological features into two research questions. As a result, the second and third research questions were formulated as follows: “RQ 2: What are the methodological features of the selected gamified FLL studies, including the research methods, data collection approaches, participants, sample size, and availability of a control group?” and “RQ 3: How has the effectiveness of gamified FLL tools been measured in previous studies?”.

Furthermore, researchers focused on different aspects of effectiveness measurement. For instance, Huang, Yang, Wang, Wu, Su and Liang [25] were interested in investigating the effectiveness of gamification on learning outcomes. Some researchers assessed the effectiveness of technology in terms of its impact on behavioral and affective engagement [26]. Dehghanzadeh, Fardanesh, Hatami, Talaee and Noroozi [26]. Behavior engagement is the task engagement that people can observe based on users’ behaviors. Indicators of behavioral engagement include time on task, task completion, attendance, and activity participation [21]. Affective engagement is the task engagement linked to learners’ emotional reactions, such as anxiety and satisfaction [21]. Since cognitive engagement is a parallel concept with behavioral and affective engagement, this author also includes it for data collection. Thus, the fourth research question was raised (RQ 4: What is the impact of gamified FLL tools on learners’ behavior engagement, affective engagement, cognitive engagement, and academic performance?).

After reviewing previous studies, such as Zainuddin, Chu, Shujahat and Perera [27] and Zou, Huang and Xie [28], this author recognized the value of underlying theoretical models for review articles. However, since the selected studies for the current research lacked a sufficient number of theoretical models, the research focus “underlying theoretical models” was not included.

Previous literature reviews also attempted to extend the current research rather than only making summaries. Focusing on digital game-based learning and vocabulary, Zou, Huang and Xie [28] summarized the main extant research issues, findings, and implications to answer the questions “where are we” and “where are we going”. The study of Zainuddin, Chu, Shujahat and Perera [27] pushed the frontier even further, as it explicitly specified one of the research questions as “what are the unexplored future research avenues in gamification research”. Summarizing or predicting future research trends is too ambitious, and thus, this author excluded it from the research questions. 

In summary, building upon previous literature reviews, the present study aimed to investigate gamification and its impact on foreign language learning in four key areas: substantive features, methodological features, measures, and effectiveness. 

## 3. Materials and Methods

This systematic literature review adhered to the typical PRISMA principles (Preferred Reporting Items for Systematic Reviews and Meta-Analyses) proposed by Moher, Liberati, Tetzlaff, Altman and Group [29]. The specific research methods utilized in this study are outlined below for transparency and clarity.

### 3.1. Search String and Databases

In the current study, four sets of keywords were utilized, including those related to gamification, effectiveness, language learning, and technological tools. As a result, the following search string was employed: (gamification OR gamified OR gamify) AND (effective OR impact) AND (language learning OR English learning OR second language OR EFL OR L2) AND (mobile application OR mobile apps OR software OR system OR apps OR platform). 

To maintain a narrow focus on gamification and language learning, certain search terms were intentionally excluded from the search string. The keyword “foreign language learning” was omitted as it overlapped with “language learning” and “second language” (abbreviated as “L2”). However, “English learning” was included due to its widespread use around the world. Additionally, terms related to video games, such as “game-based learning”, “video games”, “serious games”, and “edutainment”, were excluded. While specific aspects of language learning, such as vocabulary and writing, are theoretically important, they were also excluded due to limitations on the length of the search string.

The search string was tailored to the requirements for the word limit of different databases. The focus was on retaining the primary keywords of “gamification” and “effectiveness” while eliminating certain language-tool-related keywords. Specifically, “EFL”, “L2”, “apps”, and “platform” were removed when necessary.

To conduct a comprehensive search, this author scoured 10 academic databases, including Elsevier, Taylor & Francis Online, Sage, Wiley, Springer, JSTOR, ACM Digital Library, Scopus, ERIC, and IEEE Xplore. A total of 1752 articles were identified using the search string detailed above. These articles were expected to offer valuable insights into the effectiveness of gamification in language learning.

### 3.2. Inclusion and Exclusion Criteria

A total of 1752 articles were retrieved from the 10 academic databases using the search string described above. The titles and abstracts of these articles were carefully screened, and those that were deemed irrelevant to the current topic, not available in full text, or not written in the English language were excluded. As a result, 1688 articles were eliminated from the study.

After obtaining the 64 articles that passed the title and abstract screening, this author conducted a full-text reading, which trimmed the number to 24. Throughout this process, the exclusion criteria were carefully considered to ensure that only relevant and high-quality studies were included in the final systematic review. The exclusion criteria were as follows: -Articles that were not empirical studies that investigated the effectiveness of gamified learning tools for foreign language learning.-Duplicates were removed to ensure that each article was unique.-Empirical studies that did not properly or fully report quantitative results.-Studies that were not related to foreign language learning (e.g., heritage language and programming language).-Articles related to video games, such as “game-based learning”, “video games”, “serious games”, and “edutainment”.

To enhance the reliability of the study, a second round of full-text reading was conducted. Applying the same inclusion and exclusion criteria, this author read through the 64 articles for a second time. Three articles were subsequently removed from the study, resulting in a final selection of 21 relevant articles for the formal literature review.

### 3.3. Data Analysis

The literature review section was instrumental in shaping the research questions for the systematic literature review. As a result, the data analysis process for the study involved extracting simple answers from the selected articles. 

This author utilized frequencies to report the results of the systematic literature review. For instance, a statement such as “11 out of the 21 studies (52%) selected English as the target language” was used to present the findings.

### 3.4. Reliability and Validity

As briefed above, this author conducted a second-round full-text reading to ensure the reliability and validity of the quantitative study. Additionally, this author recognized the importance of an iterative approach to thematic analysis and reviewed the identified themes multiple times.

However, due to practical constraints, this author did not recruit a second coder for the qualitative content analysis. This represents a limitation of the current study.

## 4. Results

### 4.1. Substantive Features of FLL Gamification Studies

This section presents a summary of the substantive features of the involved empirical studies, including tool selection, target language, teaching content, and learner background (see Table 2).

It is evident that the selection of gamified learning tools varied greatly among the involved empirical studies (see Table 2). Duolingo, which is a language learning tool, was used in three studies, while a classroom response system called Kahoot! was used in two studies. The remaining studies used five unnamed tools that did not overlap. Nearly half of the studies (*n* = 10) customized the gamified learning tools for their experiments.

English was the most commonly selected language, with 11 out of the 21 studies (52%) using it as the target language. The most popular teaching objectives were vocabulary/sentence (*n* = 8) and grammar (*n* = 4). Less popular objectives included pronunciation, comprehensive language, reading comprehension, and others. Furthermore, a significant number of empirical studies did not specify the educational level for which learning was aimed (*n* = 8). One study classified their participants as “self-sufficient learners” without categorizing them by educational level [16].

### 4.2. Methodological Features of the Selected FLL-Gamification Studies

Among the 21 studies, 12 were experiments, six were quasi-experiments, two were field experiments, and one was action research (see Table 3). Table 3 also indicates that the majority of participants involved in the empirical studies were university students, adults, and primary school students. The elementary school stage was relatively understudied, with only one empirical study investigating the effectiveness of gamified FLL tools on elementary school students. The sample sizes ranged from 9 to 164 participants. Among the 21 studies, eight studies had a control group to obtain evidence on the effectiveness of gamified FLL tools, while the remaining 13 studies were conducted without control groups. 

Although previous researchers criticized the short duration of experiments in gamification studies [24,30,31], the selected articles did not yield overly pessimistic findings: if the eight non-specified cases are regarded as on-spot experiments, which is of high possibility, the number of long-term studies was still over half (11 out of 21). The data collection processes for the five studies were longer than eight weeks. The study conducted by Homer, Hew and Tan [32] intentionally allowed participants sufficient time (four months) to become familiar with the involved gamified learning tool, which helped to avoid biases resulting from novelty effects.

### 4.3. The Effectiveness of Gamified FLL Tools: Measures

While educational gamification was associated with numerous benefits, the most widely recognized advantages include the potential to increase learners’ engagement and improve academic performance [33]. Engagement, in particular, can be further divided into three components: behavioral, emotional, and cognitive engagement [34].

As such, the current study categorized measures of effectiveness assessment into five groups: behavioral engagement, emotional engagement, cognitive engagement, academic achievement, and others, with the results presented in Table 4.

According to the findings, academic achievement was the most commonly used measure to assess the effectiveness of gamified learning tools in language learning contexts. Specifically, 16 out of 21 studies (76%) employed pre- and post-tests or quizzes to collect data on academic achievement. 

Meanwhile, Table 4 indicates that emotional engagement was also a valued measure, as around half of the empirical studies included it (*n* = 10, 48%). Emotional engagement, also known as affective engagement, refers to the engagement related to emotional reactions. It encompasses various aspects such as students’ interest level, positive affect, positive attitude, positive value held, curiosity, and task absorption (and the less the anxiety, sadness, stress, and boredom) [35].

The measured emotional engagement included confidence, immersion experience, anxiety, curiosity, attitude, self-efficacy, interest, enjoyment, and general learning experience. Typically, when assessing one specific indicator of emotional engagement, researchers tended to select one validated scale from existing ones, as Chen, Li and Chen [36] did in their study (using the Game Immersion Questionnaire to measure the sense of immersion). Differently, several of the selected studies employed interviews, open-ended questions, or unvalidated questionnaires to collect data on emotional engagement [2,6,9,32,37].

Behavioral engagement refers to the observable task engagement demonstrated by users’ actions [34]. In the current study, nine of the selected articles (43%) measured behavioral engagement indicators, such as time input, number of interactions, earned points, online contributions, task completion rate, and self-reported engagement (see Table 4). Data on behavioral engagement was mainly collected through a log analysis, self-report surveys, or observations, with one study utilizing a scale specifically designed to measure behavioral engagement [38].

The lack of attention given to cognitive engagement in gamification was criticized by researchers, as it is a crucial component of effective learning [39]. Cognitive engagement refers to the psychological state in which learners are motivated to devote time and effort to fully comprehend a topic and are able to persist studying over a long period of time [40]. The present study’s findings align with those of Appleton, Christenson, Kim and Reschly [39], as only one of the gamified FLL studies measured cognitive engagement, whereas ten studies measured affective engagement and nine measured behavioral engagement.

The selected studies also included other measures too, including motivation, academic efficacy, usability, and enthusiasm, as shown in Table 4.

### 4.4. The Impact of Gamification on Foreign Language Learning

The majority of selected studies that investigated the effectiveness of gamified learning tools in foreign language learning reported positive results, as shown in Table 4. However, it is worth noting that non-positive results are even more noteworthy, as they have the potential to provide valuable insights.

Sun and Hsieh [38] aimed to investigate whether a gamified interactive response system can improve EFL learners’ motivation, engagement, and attention. Their findings showed that the gamified learning tool effectively stimulated interest, intrinsic motivation, attention, and emotional engagement, but did not show significant differences in behavioral and cognitive engagement. The reasons behind these results require further exploration.

Dindar, Ren and Järvenoja [37] conducted an experimental study on the effects of gamified cooperation and competition elements on English vocabulary learning. They found no differences in behavioral engagement (task effort), academic achievement, and motivation. However, the study did not include an un-gamified control group, and thus, the effectiveness results should be interpreted with caution.

Reynolds and Taylor [4] reported mixed results when examining the impact of a gamified EFL tool (Kahoot!) on user experiences and vocabulary knowledge. Weaknesses of the involved gamified learning tool were investigated, among which an important one was the incompatibility of gamification to some students: while the majority of students presented positive reactions, several others were demotivated in the experiment. Moreover, even though the instructors generally held a positive attitude toward the gamified learning tool, they were not entirely convinced of its efficacy related to instructions [4].

Chen, Li and Chen [36] investigated the impact of a web-based collaborative reading annotation system on reading performance. The results showed that although the experimental group generated more annotations, there was no difference in reading comprehension performance. It can be inferred that behavioral investment is not equivalent to improvements in academic achievement. 

Regarding examining other indicators, such as academic efficacy, there was also a study that reported no differences [35,41].

## 5. Discussion

### 5.1. Substantive and Methodological Features of Previous FLL-Gamification Studies

The findings of the study suggest that many of the gamified FLL tools used in the selected studies were customized (about half, *n* = 10). This was likely due to the need for greater control over variables in academic experiments, which is easier to achieve with customized tools than with commercially available ones. Moreover, the use of customized tools indicates that the gamification studies sought to advance the field by designing and testing new tools rather than simply evaluating the effectiveness of existing commercial tools. Further research is needed to explore the effectiveness and potential benefits of these customized tools in different learning contexts and with different learner groups. User studies that examine learners’ experiences with gamified FLL tools may also provide valuable insights into their effectiveness.

The use of existing commercial gamified learning tools is also promising. Findings reported that the popular selections were Duolingo and Kahoot! Duolingo, which is a typical gamified FLL tool that serves independent learning, was mainly employed in out-of-classroom time. Future studies could either explore diverse means to gamify foreign language learning or investigate teachers’ role in this type of language learning mode in which teachers are normally marginalized. Kahoot! is an interactive response system that can be used for language learning outside of classroom time. The use of Kahoot! evidenced that the gamification of language is not limited to gamifying the content; instead, it can be the gamification of the learning process. How to gamify the learning process and whether the implication can be generalized in other contexts should be explored in future studies.

It is notable that though empirical studies recruited participants from specific groups of people (e.g., university students), the empirical studies targeted self-directed learners without emphasizing their educational level [10,42]. This can inspire future studies to categorize participants based on whether they are self-directed learners or not. 

Quasi-experiment was the approach that was specifically nominated in empirical studies. Unlike true experiments or randomized controlled experiments, quasi-experiments lack random assignments to control variables, which may provide more insights into the genuine characteristics of the target population [43]. The quasi-experimental research method is necessary for ex-ante impact evaluations, or to identify general trends for future studies.

In the future, researchers could conduct more quasi-experiments to enrich the understanding of the use of gamified tools for foreign language learning or conduct true experiments based on the findings of the quasi-experiments to draw more concrete conclusions.

### 5.2. The Effectiveness Measurements of Previous FLL-Gamification Studies

It was found that using academic achievement as a sole indicator to measure the effectiveness of a gamified FLL tool may be problematic due to the influence of various factors. Thus, further validation is needed to determine the rationality of using academic achievement as a measure.

Research findings revealed that a portion of previously involved gamified FLL tools is poor at integrating gamification features with pedagogical content. Rego [3] criticized the fact that in some trials, learners must already have some language knowledge to participate in the activity, which makes the process non-educational. Similarly, in some cases, the researchers are providing “chocolate-covered broccoli” by simply adding game elements (e.g., colorful graphics and animation) to dull and repetitive tasks [2], which makes gamification essentially not very different from those of the traditional educational settings consisting of blackboards and textbooks [44].

One solution is to focus on educational content rather than game elements. It was suggested that gamification should be a holistic, creative and structured process [9] or a good and careful design on learning materials [31]. It was not specified how to gamify the learning process or learning materials, which should be addressed in future studies. Another solution is to understand how game elements should be selected, deployed, implemented, and integrated to provide a gameful experience in pedagogical practices [45]. However, it is still not clear how to push the frontier further to achieve meaningful gamification.

Previous studies presented incomparability issues since they involved varied game elements in different contexts for distinguished purposes [37]. Accordingly, future studies can explore the impact of an individual game element, as well as filter out the unsuitable game elements for educational purposes.

Researchers also concluded that gamification was considered positive when specific requirements are met [9], and thus, more efforts are expected to explore what specific requirements are needed for a successful gamification implementation.

Since the extant gamified FLL tools were criticized for laying too much emphasis on vocabulary and translation only, future tool designs are suggested to focus on other learning objectives, such as collaboration, context exploration, and reading comprehension [3]. Another aspect to consider can be social interactions, namely, the social aspects of motivation [9,17]. In assessing effectiveness, there is a need to develop the measurement to test the psychological process [37].

More aspects were highlighted in previous studies, which can also be valued in future studies. The research interests include the following: how gamification strategies improve language learning [3], how individual game elements function in the learning process [17], how gamification triggers behavior change in the learning process [3], negative impacts of gamification on students [8], how to gamify the learning process rather than simply adding game elements [45], how users perceive the use of gamified learning tools [46], and when and how to use gamification for the optimal effectiveness [47].

### 5.3. Factors Influencing the Effectiveness of Gamified FLL Tools

In the related FLL-gamification studies, researchers analyzed the contributing factors of varied effectiveness. The factors include methodological limitations, experimental bias, technical limitations, individual differences, failure in achieving meaningful gamification, a mixture of element selection, sub-optimal measurement, and data interpretation biases (see Table 5).

**There were methodological limitations.** The effectiveness of FLL-gamification studies is limited by several methodological factors such as the absence of control groups [37], inadequate control of variables [17,48], limited duration of investigation [31,49], and small sample size [17,37,50,51]. For instance, Murad, Wang, Turnbull and Wang [51] acknowledged that the major limitation of their work is that they conducted a small and short-term pilot study, and Purgina, Mozgovoy and Blake [2] noted that the experiment is hard to perform because the sample size is limited.

Dindar, Ren and Järvenoja [37] highlighted the issue of a ceiling effect in academic achievement assessments. Specifically, when a test is too easy, both the experimental and control groups tend to obtain high scores, which can obscure differences between them. Furthermore, researchers have criticized improper assessment designs.

In addition, inadequate assessment designs came under criticism from researchers. For example, Dindar, Ren and Järvenoja [37] pointed out they measured learning achievement with a vocabulary test rather than a comprehensive one that covered other language learning abilities, such as reading and writing; similarly, Lam, Hew and Chiu [52] reviewed the research process and inferred a possible sub-optimal research design: their assessment emphasized measuring one aspect of language skills, with another important one being neglected.

**There were experimental biases.** Similar to but different from methodological limitations, the experimental setting brings biases. As specified above, the lack of longitudinal studies investigating the long-term impact of gamification on foreign language learning can lead to the experimental group generating abnormal or deceptive data due to the novelty effect [30,31]. Even if the results are statistically significant, the question of whether the impact can be maintained remains an issue [53]. Other problems were reported, such as the provision of an inauthentic experience in experimental settings [1,17] and the participants’ involuntariness in learning the target language despite giving their consent to take part in the experiment [10].

**There were technical limitations.** Technical limitations have also constrained the potential of gamified foreign language learning (FLL) tools. Rego [3] proposed program developers do not always provide learners with a tool that meets their needs, either due to a lack of understanding of students’ needs or due to technical limitations. Similarly, it was observed that gamified tools can be designed too simply to induce a genuine gamification experience in empirical studies [17], or too immature in the development of dedicated language processing technologies [2].

**Individual differences were frequently cited as factors influencing effectiveness**, such as different preferences, gender, and language proficiency levels. Castañeda and Cho [6] proposed that not all students are comfortable with or suitable for the gamification approach. Alternatively, and even paradoxically, the advantages for a group of learners can be disadvantages for others. Similarly, previous studies revealed that the assumed-to-be-engaging gamified design can be frustrating or distracting to certain groups of students [31,52]. Investigating the impact of the individual game element “badges” on students, Morris, Dragovich, Todaro, Balci and Dalton [17] found that badges are more effective for learners with specific motivational classifications while undermining the motivation of more motivated students. Morris, Dragovich, Todaro, Balci and Dalton [17] concluded that it is important to identify who would benefit from gamification and who would not, thereby emphasizing the significance of considering learner differences and preferences. Gender was also identified as a contributing factor in the current study [1,54]. Unsurprisingly, gamified FLL tools have different impacts on students with varying language proficiency levels [37]. Therefore, in assessing the effectiveness of gamified FLL tools, researchers are recommended to consider the learners’ language proficiency as a variable. 

**The gamification concept was misunderstood or misinterpreted. Previous** studies showed a tendency to misunderstand or misinterpret the gamification concept. To be specific, researchers used video games in their studies despite the keyword being “gamification”. For example, Tamtama, Suryanto and Suyoto [55] specified that they used the gamification method that uses a mobile phone-based application with video games, which confuses the two concepts. Similarly, Palomo-Duarte, Berns, Dodero and Cejas [42] referred to a gamification tool as a game, even though the gamification concept is defined as using game design elements in non-game contexts.

Moreover, researchers used “pointsification” rather than gamification. Chou [56] stressed that gamification is not the simple addition of game elements; instead, it should be an engaging process that motivates learners by meeting their innate psychological needs (e.g., to gain social influence and avoid failure). However, some previous studies simply added game elements into conventional learning activities [54,57,58]. This approach, known as “pointsification”, typically involves adding points-related elements, such as badges, points, and leaderboards, to non-game contexts [59]. Although this approach technically fits the definition of gamification, it was criticized for being too simplistic and failing to provide a meaningful learning experience [54]. Thereby, pointsificaiton is an obvious factor constraining the potential of gamified FLL tools.

**Researchers used different or unsuitable game elements in varied contexts.** Werbach [45] highlighted the challenges of inherent comparability of empirical studies: even though Deterding, Dixon, Khaled and Nacke [60] defined gamification as the use of game design elements in non-game contexts, there is no universal list of game elements [45]; therefore, different studies would test different game elements with different definitions in different educational contexts, which makes it like comparing apples with bananas. Researchers also proposed that certain game elements might not suitable for pedagogical activities. For example, Abrams and Walsh [61] complained that some game design characteristics (e.g., shooting) distracted learners from vocabulary learning, and Kurniawan, Sitohang and Rukmono [31] found that three elements only brought limited effects to language learning (rules, story, and avatar). In summary, the use of different game elements raises comparability issues, and the use of unsuitable game elements may result in mixed or even opposite results.

**There was an issue of sub-optimal measurement.** There was an issue with sub-optimal measurement in assessing the effectiveness of studies. In particular, researchers often measure participants’ behavioral engagement, such as the number of interactions and time spent on specific activities, but this may not necessarily reflect the desired learning outcomes [36]. Morris, Dragovich, Todaro, Balci and Dalton [17] also noted that behavioral engagement, such as the number of earned badges, may simply represent effort input rather than cognitive improvement in learning activities. It is important to note that these two concepts have not been shown to be equivalent or identical.

To more accurately assess the effectiveness of gamified FLL tools, it is recommended that researchers consider a wider range of measurements beyond behavioral engagement. These additional measurements may include users’ psychological processes [37], users’ psychological characteristics [36], fun learning experience [58], and benefits of kinesthetic activities [4].

**There were data interpretation biases.** Data interpretation biases were identified in some studies. Murad, Wang, Turnbull and Wang [51] studied user experience by asking participants to select “agree”, “neutral”, or “disagree” to questions. When asked whether the tool was enjoyable or easy to use, none of the participants ticked “disagree” (0%); notably, when asked whether the gamified tool was helpful for vocabulary acquisition or pronunciation improvement, there was a significant number of participants selecting “neutral” or “disagree” (46.6% and 33.4%, respectively). It can be inferred that the provided gamified FLL tool could be emotionally engaging but less sufficient in bringing academic benefits. However, the authors arrived at the opposite conclusion: the selected gamified FLL tool helped participants improve pronunciation and vocabulary [51]. In particular, the authors did not stress that a considerable number of participants (33.4%) were not optimistic about the tool’s impact on pronunciation improvement. Without analyzing negative cases, the interpretation can be biased.

## 6. Conclusions

This systematic literature review aimed to identify the substantive and methodological features of the selected articles, as well as the effectiveness and measurement of the gamified FLL tools. Findings showed that the effectiveness of gamified tools in foreign language learning is mixed: they can bring positive changes, negative changes, or no difference. The factors that influenced the effectiveness could be methodological limitations, biases bought by the experiment setting, technical limitations, individual differences, failure in achieving meaningful gamification, a mixture in element selection, sub-optimal measurement, or data interpretation biases.

Three major limitations of this study were identified: the over-reliance on frequency reports in quantitative data analysis, the lack of a second coder in the qualitative data analysis, and a simple content analysis coding process that was too simplistic to be accurately replicated.

Suggestions for future research are listed in the Discussion section, such as exploring how to gamify the learning process, assessing the impact of individual game elements, and considering cognitive engagement in assessing the effectiveness of gamified FLL tools. Addressing the limitations of the current study, future ones can consider applying more complicated methods in analyzing the literature or recruiting more coders for the content analysis.

## Figures and Tables

**Table 1 behavsci-13-00331-t001:** Terms and definitions.

Term	Definition
Gamification	The use of game design elements in non-game contexts [18].
Gamified learning	The use of game design elements for educational purposes [21].
Gamified learning tool	Educational website, information system, or mobile application (mobile app) that employs game design elements [19].
Gamified FLL tool	Website, information system, or mobile application (mobile app) that employs game design elements for foreign language learning.

**Table 2 behavsci-13-00331-t002:** Substantive features of the selected empirical studies.

Theme	Result
Tool selection	Customized tools: 10. TipOn, Guess it!, SLIONS, Shadowingu, WCRAS, WordBrisks, and unnamed tools (4); Existing tools: 9. Duolingo (3), Kahoot! (2), Edmodo, Baicizhan, ClassDojo, and a gamified IRS (unnamed); Not specified: 2. “Conjugation Nation” and “Do You Speak English” (a quiz-based web application).
Language	English (11), Chinese (3), Spanish (2), bilingual—English and Chinese (1), German (1), Japanese (1), Māori (1), and Turkish (1).
Learning content	Vocabulary and sentence (8), grammar (4), comprehensive language (2), pronunciation (2), reading comprehension (1), argument writing (1), and not specified (3).
Educational level	Secondary (4), university (3), elementary (3), primary (1), self-directed learning (2), and not specified (8).

**Table 3 behavsci-13-00331-t003:** Methodological features of the selected empirical studies.

Theme	Result
Method	Experiment (12), quasi-experiment (6), field experiment (2), and action research (1).
Participants	Adult (4), elementary (1), n/a (3), primary (4), secondary (3), university (5), and other (1) (14–22 years, selected from international students from a university).
Sample size	164, 120, 120, 118, 96, 80, 75, 55, 43, 40, 30, 23, 21, 21, 20, 15, 9, and others.
Variable control	With a control group (8) and without a control group (13).
Duration	On-spot: 1–2 h (2) and n/a (8); Longitudinal study: 1 week (1), 2–4 weeks (3), 5–8 weeks (2), and more than 8 weeks (5).

**Table 4 behavsci-13-00331-t004:** Effectiveness assessment and results reported in the selected empirical studies.

Domain	Theme	Instrument	N	Result
Behavioral engagement	Time input, number of interactions, earned points, online contributions, task completion rate, and surveyed behavioral engagement	Log analysis, self-report survey, scale, and observation	9	Positive (4), no difference (1), not statistically significant (1), and n/a (3)
Affective engagement	Confidence, immersion experience, anxiety, curiosity, attitude, interest, and enjoyment General learning experience (fun, immersion, usability experience) Surveyed emotional engagement	Open-ended survey report, scale, and interview	10	Positive (8) and no difference (2)
Cognitive engagement	Surveyed cognitive engagement	Scale	1	Not statistically significant (1)
Academic achievement	Pre- and post-test	Test and quiz	16	Positive (11), no difference (2), and no significant difference (1)
Others	Motivation, academic efficacy, attention, usability, and enthusiasm	Observation, survey, and self-report journal	6	Not found or worse (1) and partially positive (1)

**Table 5 behavsci-13-00331-t005:** Contributing factors that influence the effectiveness of gamified FLL tools.

Domain	Factors
Methodological limitations	The absence of control groups
	Inadequate control of variables
	Limited duration of investigations
	Small sample size
	Ceiling effects of easy tests
	Inadequate assessment designs
Biases bought about by experimental settings	Novelty effect
	Inauthentic learning experience in experimental settings
	Involuntariness in taking part in certain activities
Technical limitations	Designs that do not meet needs
	Simple designs that are unable to provide genuine gamification experiences
	Immature techniques in processing language-related issues
Individual differences	Learner differences or learner preferences
	Gender
	Language proficiency
Failure to achieve meaningful gamification	The involvement of video games in the name of gamification
	The involvement of conventional educational activities in the name of education (e.g., quizzes)
	The use of “pointsification” as gamification
A mixture in element selection	Comparability issue: the use of different game elements in different contexts
	The use of unsuitable game elements in educational activities
Sub-optimal measurement	A lack of measurement of the psychological process
	A lack of measurement of psychological characteristics
	A lack of measurement of fun learning experience
	A lack of measurement of the benefits of kinaesthetic activities
Data interpretation biases	A lack of negative case analysis

## Data Availability

The datasets generated and analyzed during the current study were uploaded to Figshare: https://doi.org/10.6084/m9.figshare.20317293.v1 (accessed on 5 April 2023).

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
