# Peer review of "The Effectiveness of Gamified Tools for Foreign Language Learning (FLL): A Systematic Review"

_behavsci, 2023, doi:10.3390/bs13040331_

Round 1

Reviewer 1 Report

- The abstract is clear and to the point

- The introduction presents an interesting and topical topic; it is informative and the gap in the literature is identified. Clear RQs.

- Literature Review: clear, important and basic information is given.

- Methodology: some choices are justified well but some others are not; academic sources are needed and this is a major weakness of the paper. Check comments in the paper.

- Results are clear and tables are helpful.

- Discussion: sub-sections 5.1 and 5.2 are rather descriptive with weak critical analysis and comparisons. These two need attention and maybe consider rewriting them. The rest of the sub-sections are discussed well but have some points that need to be further explained and expanded (check comments in the paper). Also, there is some wrogn numebring here, e.g. 5. Discussion, 6.1. Substantive and methodological features...

- There are a lot issues with the use of English: weak sentence conenxtion, no use of linking words and relative pronouns, spelling mistakes, grammar mistakes, syntax mistakes, very abrupt transitions and some non-acacamic language (check comments in the paper).

- Clear conclusions.

- Interesting, up-to-date, relevant and a range of sources is used.

Author Response

Comment 1#1: The abstract is clear and to the point. The introduction presents an interesting and topical topic; it is informative and the gap in the literature is identified. Clear RQs. Literature Review: clear, important and basic information is given. Results are clear and tables are helpful. Clear conclusions. Interesting, up-to-date, relevant and a range of sources is used.

Author response: Thank you.

Comment 1#2: Methodology: some choices are justified well but some others are not; academic sources are needed and this is a major weakness of the paper. Check comments in the paper.

Author response: This systematic literature review adheres to the typical PRISMA principles (Preferred Reporting Items for Systematic Reviews and Meta-Analyses) proposed by Moher et al. (2009).

Comment 1#3: Discussion: sub-sections 5.1 and 5.2 are rather descriptive with weak critical analysis and comparisons. These two need attention and maybe consider rewriting them. The rest of the sub-sections are discussed well but have some points that need to be further explained and expanded (check comments in the paper). Also, there is some wrong numebring here, e.g. 5. Discussion, 6.1. Substantive and methodological features...

Author response: Revised as suggested. Thank you.

Comment 1#4: There are a lot issues with the use of English: weak sentence connection, no use of linking words and relative pronouns, spelling mistakes, grammar mistakes, syntax mistakes, very abrupt transitions and some non-academic language (check comments in the paper).

Author response: I have sent the manuscript to a professional proofreader. I believe the current manuscript is much better. Please kindly check the revised manuscript. Thank you very much.

Comment 1#5: Try to avoid such short sentences and connect them smoothly to the previous ones. There are several such cases below and such short sentences may could meaning.

Author response: Revised as suggested.

Comment 1#6: “By screening titles and abstracts, this author removed articles that are not aligned with the current topic, not available in full-texts, and not written in English language”- Were these criteria created? Any sources to support this decision and criteria?

Author response: It has been revised as “The titles and abstracts of these articles were carefully screened, and those that were deemed irrelevant to the current topic, not available in full-text, or not written in the English language were excluded. As a result, 1688 articles were eliminated from the study.”

Comment 1#7: “There are individual differences. Individual difference was frequently mentioned as a factor influencing effectiveness, such as different preferences, gender, and language proficiency levels.” Explain and expand on this.

Author response: Revised as suggested. Thanks for the comment.

Reviewer 2 Report

This is very clearly written literature review. 

I have few comments regarding the structure of some subsections. Please see the annotated pdf. 

Author Response

Comment 2#1: This is very clearly written literature review. 

Author response: Thank you very much.

Comment 2#2: I would advise the author to add the refer gaps passage to the previous passage.

Author response: Revised as suggested.

Comment 2#3: The research questions should be moved after the literature review. This will make it for the readers easier to follow. 

Author response: Revised as suggested.

Comment 2#4: Change “Rationale of RQ selection”.

Author response: Revised as suggested. It has been changed to “Research focuses of other literature reviews”.

Round 2

Reviewer 1 Report

- Clear abstract and Introduction

- Comprehensive literature review. I like how the RQs are within the literature review and their connection canb clearly identified.

- Methodology is clearer and more robust

- Very interetsing and clear results and discussion

- Very interetsing and up-to-date sources

- Some issues with too many very short paragraphs; consider merging them to boost cohesion and coherence of the paper. This is evident throughout the paper. Check the comment in the paper.

Author Response

Comment 1#1: The sentence “Modifications were made” is a bit of abrupt. Consider rephrasing.

Author response: Revised as suggested. That paragraph has been changed to “After reviewing previous studies, such as Zainuddin et al. (2020) and Zou et al. (2019), this author recognized the value of underlying theoretical models for review articles. However, since the selected studies for the current research lacked a sufficient number of theoretical models, the research focus ‘underlying theoretical models’ was not included.”

Comment 1#2: I think this is not needed, since the RQs are smoothly connected and incorporated in the literature review.

Author response: I appreciate your feedback. I believe that having a dedicated section for research questions will make it easier for readers to locate and understand the specific objectives of this study. Therefore, if possible, I would like to keep the section for research questions. Thank you very much.

Comment 1#3: On page 5 there are too many extremity short paragraphs that discuss the same ideas. I suggest they should be merged into one paragraph. Consider this for other similar cases.

Author response: I have revised it as suggested. The three paragraphs were combined into one: “To maintain a narrow focus on gamification and language learning, certain search terms were intentionally excluded from the search string. The keyword "foreign language learning" was omitted as it overlapped with "language learning" and "second language" (abbreviated as "L2"). However, "English learning" was included due to its widespread use around the world. Additionally, terms related to video games, such as "game-based learning," "video games," "serious games," and "edutainment," were excluded. While specific aspects of language learning, such as vocabulary and writing, are theoretically important, they were also excluded due to limitations on the length of the search string.”